# Automatic Fabric Defect Detection Method Using AC-YOLOv5

Yongbin Guo [1], Xinjian Kang [1], Junfeng Li [1,2,*] and Yuanxun Yang [1]

1 School of Information Science and Engineering, Zhejiang Sci-Tech University, Hangzhou 310018, China; gybzjlg@163.com (Y.G.); 202230705139@mails.zstu.edu.cn (X.K.); yyx1600700937@outlook.com (Y.Y.)
2 Tongxiang Research Institute, Zhejiang Sci-Tech University, Tongxiang 345000, China
* Correspondence: ljf2003@zstu.edu.cn

**Abstract:** In the face of detection problems posed by complex textile texture backgrounds, different sizes, and different types of defects, commonly used object detection networks have limitations in handling target sizes. Furthermore, their stability and anti-jamming capabilities are relatively weak. Therefore, when the target types are more diverse, false detections or missed detections are likely to occur. In order to meet the stringent requirements of textile defect detection, we propose a novel AC-YOLOv5-based textile defect detection method. This method fully considers the optical properties, texture distribution, imaging properties, and detection requirements specific to textiles. First, the Atrous Spatial Pyramid Pooling (ASPP) module is introduced into the YOLOv5 backbone network, and the feature map is pooled using convolution cores with different expansion rates. Multiscale feature information is obtained from feature maps of different receptive fields, which improves the detection of defects of different sizes without changing the resolution of the input image. Secondly, a convolution squeeze-and-excitation (CSE) channel attention module is proposed, and the CSE module is introduced into the YOLOv5 backbone network. The weights of each feature channel are obtained through self-learning to further improve the defect detection and anti-jamming capability. Finally, a large number of fabric images were collected using an inspection system built on a circular knitting machine at an industrial site, and a large number of experiments were conducted using a self-built fabric defect dataset. The experimental results showed that AC-YOLOv5 can achieve an overall detection accuracy of 99.1% for fabric defect datasets, satisfying the requirements for applications in industrial areas.

**Keywords:** fabric defect; surface defect detection; deep learning; attention mechanism



## 1. Introduction

During the manufacturing process of textiles, various factors, such as the limitations of textile machinery, human error, and material quality. can cause defects in the fabric, such as broken yarns, misalignments, holes, and snags. If these defects are not detected and corrected in a timely manner, they can lead to a reduction in production efficiency and product quality, resulting in significant waste. As a result, performing textile defect detection can improve production efficiency, product quality, reduce production costs and boost the textile industry.

In the traditional textile industry, defect detection has always relied on manual and visual inspection. However, manual detection is prone to subjective judgments and is time-consuming and expensive for large-scale production. Traditional visual methods are also limited in handling non-structured and highly variable imperfections, and they lack flexibility and adaptability when faced with large amounts of production data processing. With the development of machine vision and deep learning, automated detection techniques have become feasible in the industry. However, the main challenge of automatic detection techniques is to address the problem of high false alarm rates and missed detection rates to improve the accuracy and stability of textile defect detection.

Currently, deep learning and machine vision have been applied in various fields. Machine vision-based defect detection methods primarily include those based on statistical analysis [1], frequency–domain analysis [2], model-based analysis [3,4] and machine learning [5]. Deep learning has strong feature expression, generalization, and cross-scene capabilities. With the development of deep learning technology, defect detection methods based on deep learning have become widely used in various industrial scenarios, particularly in solar energy [6], liquid crystal panels [7], railway transportation [8], metal materials [9], and other fields.

There are relatively high requirements for the detection of fabric defects, most of which tend to be broken warp, broken weft, warp shrinkage, weft shrinkage, torn holes, loose warp, and loose weft under 100 microns. Additionally, the exact location of the defect must be marked to optimize the production process and equipment parameters. A deep learning classification network [10] can only obtain the coarse positioning of the target, the positioning accuracy is related to the size of the sliding window and the classification performance of the network, and the speed is also relatively slow. The target detection network [11] is the closest network to the defect detection task, and it can obtain the accurate location and classification information of the target at the same time. The object detection network is generally divided into a single stage and two stages. The two-stage network first obtains bounding boxes based on the location of the discovered target object to ensure sufficient accuracy and recall, then it finds a more accurate location by classifying the bounding boxes. Two-stage algorithms have high accuracy but slow speed, and include R-CNN [12], SPP-Net [13], FastR-CNN [14], and FasterR-CNN [15]. Instead of obtaining bounding boxes, the single-stage network directly generates the categorical probabilities and position coordinate values of the objects. The final detection result can be directly obtained through a single detection. The speed of single-phase networks, which include SSD and the YOLOv3 [16], YOLOv4 [17], YOLOv5 [18], YOLOv6 [19], and YOLOv7 [20] series, is generally faster than the two-stage network speed, but there is a small loss of accuracy.

YOLOv5 is a single-stage object detection network with excellent performance, enabling end-to-end training without interference from intermediate processes and a fast detection speed that can meet the requirements of real-time fabric detection. However, fabric texture backgrounds are complex, with different sizes and types of defects. The features of some minor defects are highly similar to the background information and are difficult to distinguish with the human eye. Direct application of YOLOv5 to fabric defect detection poses a significant challenge. Taking into account the optical properties, texture distribution, defect imaging characteristics, and detection requirements of textiles, therefore this paper proposes a YOLOv5 defect detection network based on atrous spatial pyramid pooling (ASPP) and an improved channel attention mechanism. An automatic detection system for fabric defects is developed, and its industrial application is achieved.

The remainder of this paper is organized as follows: Section 2 presents related work. Section 3 presents the fabric defect detection system. Section 4 details the detection method, including the network structure and loss function of AC-YOLOv5. Section 5 presents the experimental validation of our method. Section 6 concludes our work and discusses the advantages and disadvantages of AC-YOLOv5 and related future research.

The primary contributions of this study are as follows:

(1) The ASPP module is introduced into the YOLOv5 backbone network. This module constructs convolution kernels for different receptive fields with different dilation rates to obtain multiscale object information. When performing feature extraction on images, it has a large receptive field. At the same time, the resolution of the feature maps does not significantly decrease, which greatly improves the fabric defect detection capability of the YOLOv5 network.

(2) A CSE attention mechanism is proposed, wherein a convolutional channel is added to the SE network and the sum of the two outputs is taken as the result of the CSE module. The introduction of the CSE module into the YOLOv5 backbone network can enhance the large defect detection capability.

(3) Combined with the CSE and ASPP modules, we propose a modified YOLOv5 defect detection network. With an average detection accuracy of 99.1%, we have achieved automatic, accurate, and robust detection of fabric defects.

## 2. Related Work

### 2.1. Fabric Defect Detection Based on Machine Vision

Liu and Zheng [21] proposed an unsupervised fabric defect detection method based on the human visual attention mechanism. The two-dimensional entropy associated with image information and texture is used to model the human visual attention mechanism, then the quaternion matrix is used to reconstruct the image. Finally, the quaternion matrix is transformed into the frequency domain using the hypercomplex Fourier transform method. Experiments have shown that the proposed method performs well in terms of accuracy and adaptability, but the time cost due to matrix operations still requires optimization. Additionally, the method cannot be used for defect detection in fabrics with periodic patterns. Jia [22] proposed a new fabric defect automatic detection method based on lattice segmentation and template statistics (LSTS). This approach attempts to infer the placement rules of texture primitives by partitioning the image into non-overlapping lattices. The lattices are then used as texture primitives to represent a given image with hundreds of primitives instead of millions of pixels. However, the time requirement of the lattice partitioning is different for different patterns. Additional template data comparisons may also slow down the run in order to improve accuracy. Song [23] proposed an improved fabric defect detection method based on the fabric area membership (TPA) and determined the significance of the defect area by analyzing the regional characteristics of the fabric surface defects. This approach requires a large amount of feature extraction and analysis work, which is difficult and susceptible to environmental factors such as lighting conditions and the camera used.

### 2.2. Fabric Defect Detection Based on Deep Learning

Jing et al. [24] proposed a very efficient convolutional neural network, Mobile-Unet, to achieve end-to-end defect segmentation. This approach introduces deep separable convolutions, which greatly reduce the complexity cost of the network and model size. However, as a supervised learning approach, it still requires considerable human effort to label defects. Wu [25] proposed a wide and light network structure based on Faster R-CNN to detect common fabric defects and improve the feature extraction capability of the feature extraction network by designing an extended convolution module. Detection can be relatively slow when processing large-scale, high-resolution images. The design of dilated convolutional modules requires a large number of experiments and fine-tuning, which increases the time and energy cost of algorithm design. Li [26] proposed three methods—multiscale training, dimensional clustering, and soft nonmaximum suppression instead of traditional nonmaximum suppression—to improve the defect detection capability of R-CNN. This approach enlarges or reduces the detailed information, neglecting the different characteristics of the defect regions at different scales. This can lead to suboptimal detection results. The YOLO algorithm family has proven to be efficient and accurate in object detection, but there is still room for improvement. In recent years, improvements based on the YOLOv3 and YOLOv4 [27,28] algorithms have been continuously proposed. Training on multiple datasets results in improved accuracy and detection speeds.

### 2.3. Fabric Defect Detection Based on Machine Vision and Deep Learning

Chen [29] proposed a new method of two-stage training based on a genetic algorithm (GA) and backpropagation. This method leverages the advantages of the Gabor filter in frequency analysis and embeds the Gabor core into the Faster R-CNN convolution neural network model. However, the combination of genetic and backpropagation algorithms requires significant computational resources and time, which can lead to slow algorithm processing. To combine the characteristics of single-stage and two-stage networks, Xie and

Wu [30] proposed a robust fabric defect detection method based on the improved RefineDet. Using RefineDet as the basic model, this approach inherits the advantages of the two-stage detector and the first-stage detector, and can detect defective objects efficiently and quickly. However, the robustness of this approach requires validation on a large number of instance datasets. If the dataset does not cover all types of imperfections, it may lead to unstable algorithm performance.

## 3. Fabric Defect Detection System

The fabric defect visual detection device designed and developed in this paper is shown in Figure 1. It primarily includes an image acquisition system and an image processing system. The image acquisition system consisted of a 2K area array camera and multiple light sources. This system can image the fabric produced by the circular knitting machine with high quality and capture defects such as broken warp, broken weft, warp shrinkage, weft shrinkage, torn holes, loose warp, and loose weft. The image processing system consisted of an industrial computer and detection system software to achieve accurate and real-time detection of various fabric defects.

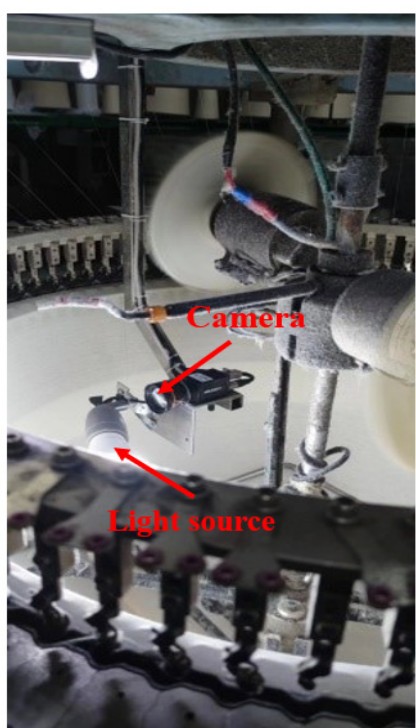
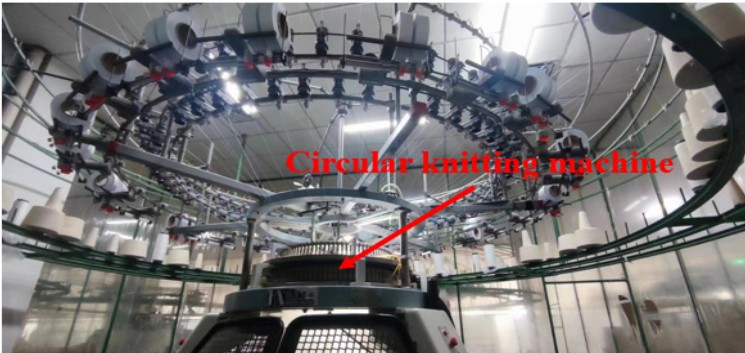
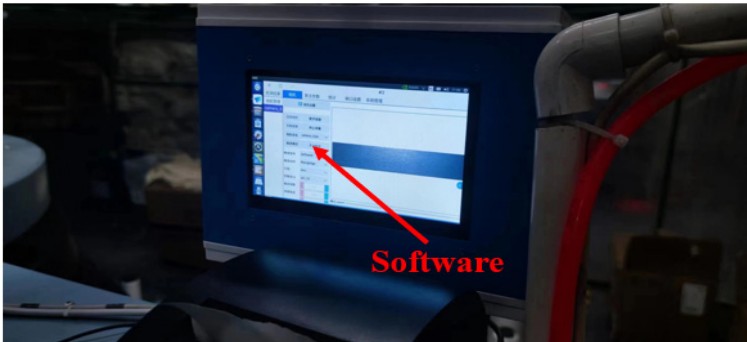

**Figure 1.** Fabric defect detection system.

Since there are many kinds of fabric defects, directly using the deep learning network to detect them will not only increase the structure of the network but also reduce the accuracy and efficiency of defect detection. Therefore, fabric defects are classified into three categories in this paper: holes, long strip (L_line) defects, and short strip (S_line) defects, as shown in Figure 2. And the different defects in the figure are marked with red boxes.

According to the imaging characteristics, texture distribution, and detection requirements of fabric defects, the difficulties of fabric defect detection are as follows:

(1) When collecting fabric images, due to the loss of three-dimensional structure information for the defects, different types of defects become very similar in appearance.

(2) The fabric defect detection has high requirements. The detection network must be able to process high-resolution images and extract feature information.

(3)   The fabric defects are complex and diverse. Although the causes of different types of defects are different, the appearance may not be different, and the size of defects in the same category may also be different.

(4)   The texture and color of fabrics are becoming increasingly diverse, and the complex backgrounds will pose significant challenges to detection.

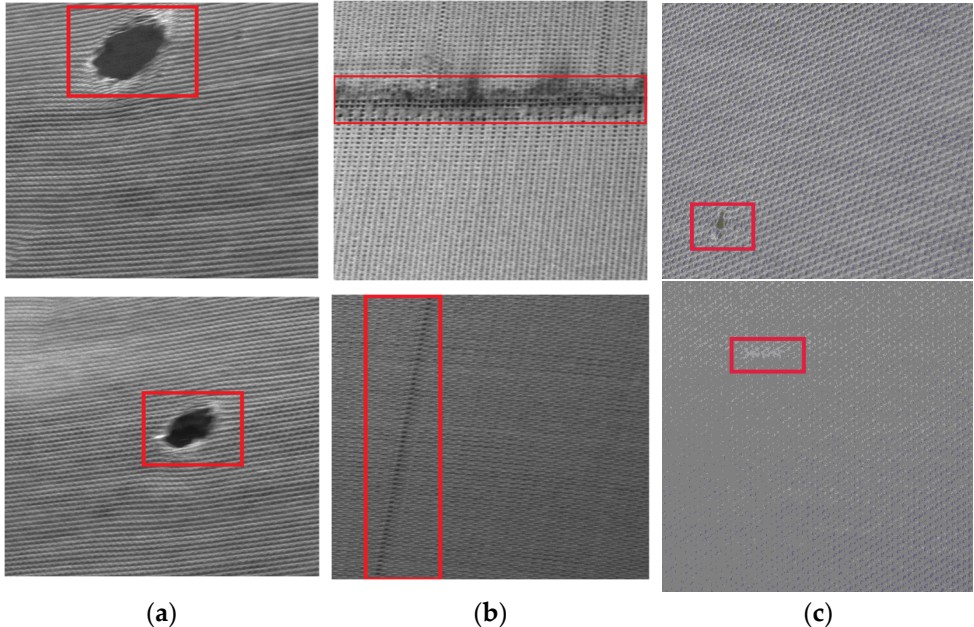

**Figure 2.** Fabric defect dataset. (**a**) Hole defects, (**b**) L_line defects, (**c**) S_line defects.

## 4. Fabric Defect Detection Method Based on AC-YOLOV5

Yolov5 is composed of a backbone network, a neck network, and a detection head. The backbone network achieves feature extraction, the neck network achieves feature fusion, and the detection head outputs prediction results. Yolov5 uses CSPDarknet53 as the backbone network. Combining the feature pyramid network (FPN) [31] and pixel aggregation network (PAN) [32] as the neck network, it is used to fuse the features extracted from the backbone network. At the same time, YOLOv5 uses a mosaic data enhancement method to splice four images by flipping, random clipping, brightness change, and other methods to enrich the image information and enhance the robustness of the network. YOLOv5 is comparable to YOLOv4 in terms of accuracy, but it is significantly faster and easier to deploy than YOLOv4. YOLOv5 is currently one of the most commonly used single-stage target detection network [33].

YOLOv5 uses a convolution kernel with a size of 3 × 3. Although the deep feature information can be extracted through multiple downsamplings, it reduces the resolution of the feature map and leads to the loss of some shallow information. Consequently, it causes difficulties in detecting small targets and is not conducive to positioning. In this paper, we propose a modified YOLOv5 defect detection network that combines various spatial pyramid pooling and channel attention mechanisms. The network structure is shown in Figure 3.

The backbone network consists of Focus, CBS, C3, SPP, and ASPP modules. The Focus module is used to convert high-resolution image information from spatial latitude to channel latitude. The CBS module consists of a convolution operation, batch normalization, and SILU activation function, which is the basic module of the backbone network. The C3 module consists of one bottleneck module, three outer CBS modules, and one concat module. In the figure, N in C3-N represents the number of stacked bottleneck modules. The design idea for the bottleneck module is inspired by the residual network to smooth the flow of positive and negative gradients. The combination design of three external CBS modules and concat modules is derived from from CSPNet [34]. The input feature map passes through two paths. One path involves a convolution of 1 × 1 followed by a bottleneck

modul, the other path goes through a CBS module, and the number of convolutional channels is reduced by half. After concatenating the output of the bottleneck module, it is adjusted to the number of output channels of the C3 module via a CBS module. The SPP module can increase the translation invariance of the network and output images of different sizes into a fixed dimension. ASPP is used to obtain multiscale information of the feature maps and enhance the information extraction capability of the backbone network.

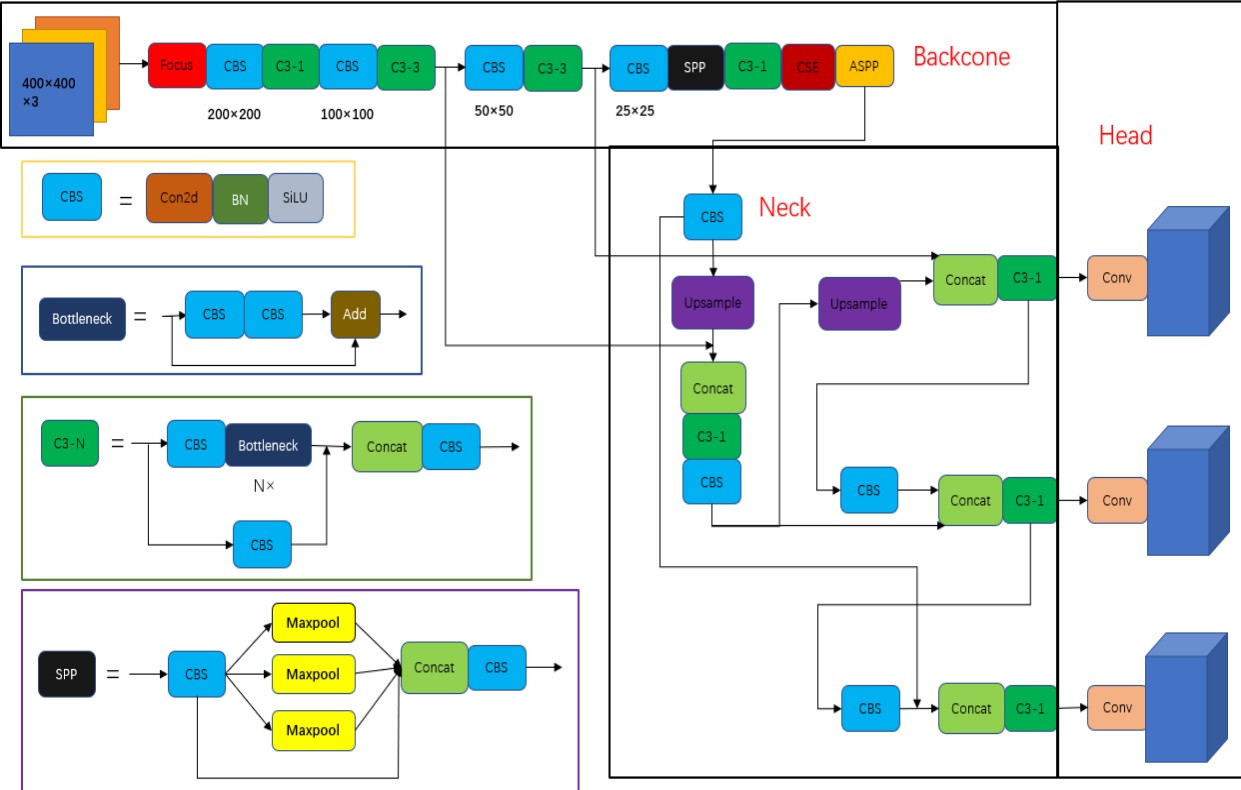

**Figure 3.** AC-YOLOV5 network structure.

The neck network fuses four layers of feature maps through four concat modules to fully extract contextual information, which reduces the loss of feature map information and improves the recognition accuracy of the network. Networks of different depths can be used to identify objects of different sizes. To adapt to changes in object size during object detection, it is necessary to fuse the feature information from different depths in the backbone network. YOLOv5 uses the FPN + PAN network. Both the FPN and PAN modules are based on the pyramid pooling operations, but in different directions. FPN facilitates the detection of large objects through top-down sampling operations. PAN improves the detection rate of small objects by transferring feature information from the bottom to the top. The combination of the two structures strengthens the feature fusion capability of the network.

In addition, to further enhance the feature extraction capability of YOLOv5, a CSE module is proposed in combination with the channel attention mechanism and convolutional module, which is introduced into the backbone network of YOLOv5 to greatly improve the feature extraction capability of the backbone network.

### 4.1. ASPP Module

The ASPP module [35] was first proposed in DeepLabv2. Although it was proposed to improve the performance of the semantic segmentation network, its method can also be used to improve the target detection network. ASPP uses convolution kernels with different expansion rates to pool the characteristic images and obtain the characteristic

images of different receptive fields to extract the characteristic information at multiple scales without increasing the number of parameters or changing the resolution of the input image. As shown in Equation (1), *r* represents the expansion rate. By adding $(r - 1)$ zeros in the middle of the original convolution core, convolution cores of different sizes can be obtained. Because zero is added, the parameters and calculations will not be increased. A value of $r = 1$ represents the standard convolution.

$$y[i] = \sum_k x[i + r \cdot k] w[k] \tag{1}$$

As shown in Figure 4, ASPP uses a convolution kernel with a size of $3 \times 3$ to extract features at four scales from the feature map through the atrous convolution kernel with expansion rates of 6, 12, 18, and 24. It obtains four feature maps of different receptive fields, which are spliced together through the concat module to achieve multiscale feature extraction.

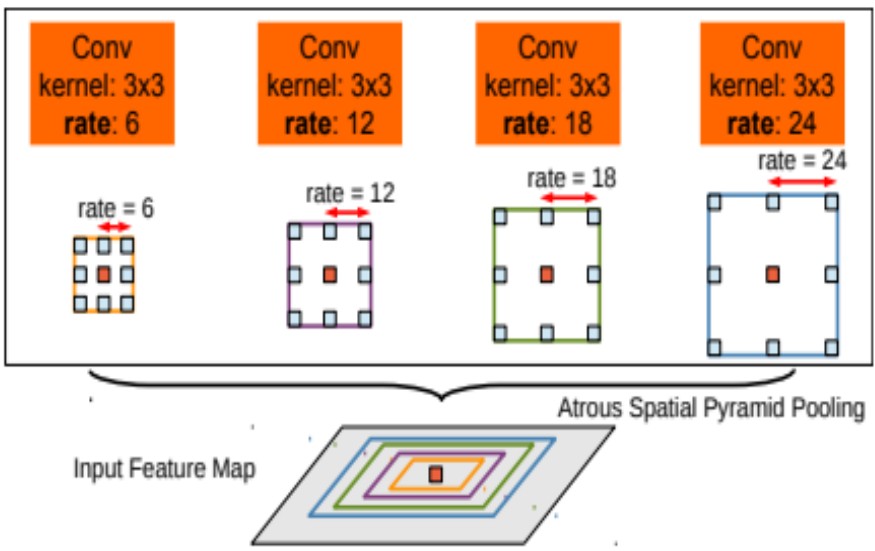

**Figure 4.** ASPP multiscale convolution.

### 4.2. CBE Module

The attention mechanism can quickly locate the target information within a large amount of information. Introducing an attention mechanism into the YOLOv5 network and assigning greater weight to the fabric defect target can make the network prioritize areas with defects and improve the network's defect detection ability. The SE network [36] can determine the importance of each feature channel through self-learning, assign corresponding weights to the channels, increase the learning of target information, and ignore some interference information. The SE network is shown in Figure 5.

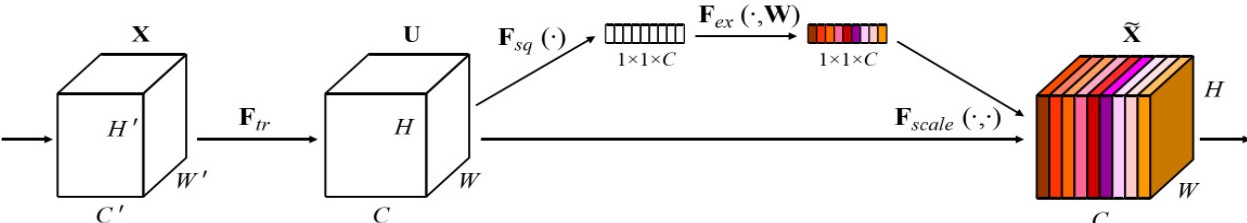

**Figure 5.** Squeeze-and-excitation networks.

The SE module is regarded as a computing unit, which establishes the convolution mapping of $F_{tr} : X \rightarrow U$, as shown in Equation (2). * represents the standard convolution operation, $X \in R^{H' \times W' \times C'}$ represents the input, $U = [u_1, u_2, \ldots, u_c] \in R^{H \times W \times C}$ represents

the output, the convolution kernel is $V = [v_1, v_2, \ldots, v_c]$, $v_c$ represents the $c^{th}$ convolution kernel, and $v_c^s$ represents the 2D convolution kernel on the $s^{th}$ channel.

$$u_c = v_c * X = \sum_{s=1}^{C'} v_c^s * x^s \tag{2}$$

$$z_c = F_{sq}(u_c) = \frac{1}{H \times W} \sum_{i=1}^{H} \sum_{j=1}^{W} u_c(i,j) \tag{3}$$

$$s = F_{ex}(z, W) = \sigma(g(z, W))\sigma(W_2 \delta(W_1 z)) \tag{4}$$

$$\widetilde{X} = Fscale(u_c, s_c) = s_c \cdot u_c \tag{5}$$

SE consists of three parts: squeeze, excitation, and scale. The specific structure is shown in Figure 6. First, global average pooling is used to compress the feature maps with a size of $W \times H \times C$ to a size of $1 \times 1 \times C$ ($C$ is the number of channels). This operation produces the vector z, as shown in Equation (3), which converts the spatial features of each channel into global features with a global receptive field. Then, the $Z$ vector is sent into the two fully connected layers and the ReLU activation function to learn the correlation of the channel. The first full connection layer reduces the parameters by reducing the number of channels, and the second full connection layer restores the dimension of the channel and normalizes the channel weight using the sigmoid function, as shown in Equation (4). Finally, the obtained weight is scaled to the characteristics of each channel, as shown in Equation (5). This process adjusts the input feature mapping using the weight to improve the sensitivity of the network to fabric defects.

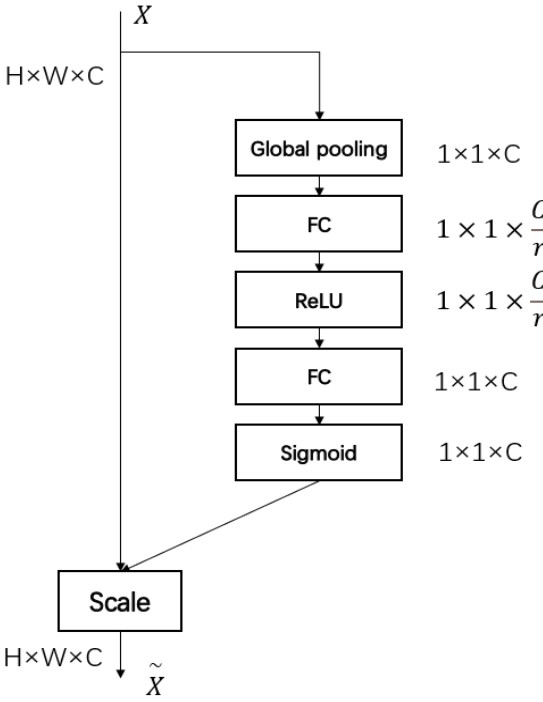

**Figure 6.** SE module structure.

The SE module improves the sensitivity of the network to the channel characteristics and is lightweight, imposing little burden on network computing. However, the SE block also has limitations. In the squeeze module, the global average pool is too simple to capture

complex global information. In the excitation module, the fully connected layer increases the complexity of the model. Based on this, a CSE module combining the SE attention mechanism and the convolution module is proposed in this paper. The CSE module can greatly improve the detection ability of the network for long and narrow defects by adding the channel-weighted results and the $3 \times 3$ convolution results. The CSE module structure is shown in Figure 7.

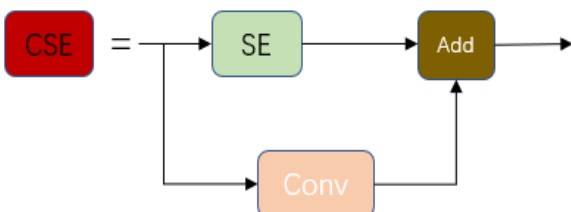

**Figure 7.** CSE module structure.

### 4.3. Loss Function

The loss function is used to measure the difference between the real tag value and the predicted value of the model. The selection of the loss function affects the network performance, and the function value is inversely proportional to the network performance. The loss function presented in this paper includes three parts: boundary box regression loss, confidence prediction loss, and category prediction loss. The total loss function is shown in Equation (6):

$$Loss = \omega_{box} L_{box} + \omega_{obj} L_{obj} + \omega_{cls} L_{cls} \tag{6}$$

where *lbox* is the positioning error function used to calculate the error of the prediction box and real box, *lobj* is the confidence loss function used to calculate the network confidence error, and *lcls* is the classification loss function used to calculate whether the classification is correct. $w\_box$, $w\_obj$ and $w\_cls$ are weight values, which are 0.05, 0.5, and 1, respectively.

The positioning error function uses the *CIOU* loss function, as shown in Equation (7):

$$L_{box} = CIOU = 1 - IOU + \frac{\rho^2(b, b^g)}{c^2} + \alpha v \tag{7}$$

$$\alpha = \frac{v}{1 - IOU + v} \tag{8}$$

$$v = \frac{4}{\pi^2} \left( arctan \frac{w^g}{h^g} - arctan \frac{w}{h} \right)^2 \tag{9}$$

where $\rho^2(b, b^g)$ represents the Euclidean distance between the center point of the prediction box and the center point of the real box, and $c$ represents the length of the minimum closed box diagonal covering the prediction box and the real box. $\alpha$ as a weight coefficient, as shown in Equation (8), $h^g$ and $w^g$ are the length and width of the prediction box, and $h$ and $w$ are the length and width of the real box.

Both the classification loss function and the confidence loss function adopt the binary cross entropy loss function, as shown in Equation (10):

$$L_{cls} = L_{obj} = -\frac{1}{n} \sum (y_n \times ln x_n + (1 - y_n) \times \ln(1 - x_n)) \tag{10}$$

where $n$ represents the number of samples entered, $y_n$ represents the true value of the target, and $x_n$ represents the predicted value of the network.

## 5. Experiment and Analysis

### 5.1. Fabric Defect Dataset

The self-built fabric defect dataset used in this paper was obtained from the production line. It was taken by the industrial area array camera, resulting in $400 \times 400$ resolution images after cutting processing. The total number of images was 2764. Skilled technicians then classified and labeled the images. Considering the difficulty of detecting different types of defects, the number of images was relatively small due to the regular shape of the holes. More images were collected for the L_Line and S_Line: 1644 and 877, respectively, as shown in Table 1.

**Table 1.** Fabric defect dataset.

|        | L_line | S_line | Hole |
|--------|--------|--------|------|
| Number | 1644   | 877    | 243  |

The dataset of the proposed AC-YOLOv5 algorithm consisted of a training set, a validation set, and a test set. Each type of defect image and label was roughly divided into 70%, 10%, and 20%. The results are shown in Table 2.

**Table 2.** Dataset training, verification, and test set division.

|        | Training | Validation | Test | Total |
|--------|----------|------------|------|-------|
| L_line | 1046     | 279        | 319  | 1644  |
| S_line | 567      | 127        | 183  | 877   |
| Hole   | 155      | 37         | 51   | 243   |

### 5.2. Software and Hardware Environment Settings

The hardware environment and software version used in this experiment are shown in Table 3, and the spectrometer setup is shown in Table 4.

**Table 3.** Hardware environment and software version.

| Hardware and Software | Configurations |
|-----------------------|----------------|
| DESKTOP-7V5KI6L | Operating System: windows 10 |
|  | CPU:Intel(R) Xeon(R) Gold 6136 |
|  | RAM:256G |
| Software version | GPU:NVIDIA Quadro RTX 6000 |
|  | Pycharm2021 + Python3.8 + CUDNN7.6.05 + Opencv4.5.2.54 + CUDA10.2 |

**Table 4.** Network training parameters.

| Training Parameters | Value |
|---------------------|-------|
| Batch size | 1 |
| Dynamic parameters | 0.937 |
| Learning rate | 0.01 |
| Cosine annealing learning rate | 0.01 |
| Data augmentation | 1.0 |
| Input image size | $400 \times 400$ |
| Epochs | 100 |

### 5.3. Performance Metrics

To quantitatively analyze the test results, three evaluation metrics are used in this paper: precision, recall, and mAP.

$$R = \frac{TP}{TP + FN} \tag{11}$$

$$P = \frac{TP}{TP + FP} \tag{12}$$

Whenever $TP$ represents a defect on the fabric with a true detection result, $FP$ represents a defect that is not on the fabric but has a true detection result, and $FN$ represents a defect that is not on the fabric but has a false detection result.

The specific meanings of $TP$, $FP$ and $FN$ are listed in Table 5:

**Table 5.** Confusion Matrix.

| Real | Prediction | |
|---|---|---|
| | **Positive** | **Negative** |
| True | $TP$ | $TN$ |
| False | $FP$ | $FN$ |

Here, "Real" represents the real defects on the fabric, and "Prediction" represents the predicted results.

The accuracy and average accuracy are calculated as follows:

$$AP = \int_0^1 P(R)dR \tag{13}$$

$$mAP = \frac{\sum AP}{N} \tag{14}$$

Here, $AP$ denotes the average detection accuracy of each category, and $N$ denotes the number of categories in the dataset.

### 5.4. Ablation Experiment

AC-YOLOv5 improves upon YOLOv5. To verify the validity of the model, ablation experiments were performed and presented in this paper. The experimental results are shown in Table 6, which shows that the mAP of the YOLOv5 network was 98.2%. After adding the ASPP module to the backbone network alone, the mAP was increased to 98.6%, and the recall was reduced. By adding the CSE module alone, the mAP was increased to 98.8%, but the detection speed was reduced. With the addition of the ASPP module and the CSE module, the detection accuracy reached 99.1%, and the detection speed did not decrease.

**Table 6.** Results of ablation experiment.

| Method | P | R | mAP | FPS | Flops |
|---|---|---|---|---|---|
| YOLOv5 | 95% | 95.1% | 98.2% | 476 | 15.8 |
| YOLOv5 + ASPP | 97.9% | 92.6% | 98.6% | 476 | 18.5 |
| YOLOv5 + CSE | 95.1% | 97.5% | 98.8% | 454 | 17.7 |
| AC-YOLOv5 | 97.8% | 98.5% | 99.1% | 476 | 20.4 |

### 5.5. Comparative Experiment

To verify the effectiveness of the proposed model, the AC-YOLOv5 network was compared with other common object detection networks. The comparison results are presented in Table 7, which shows that AC-YOLOv5 had the highest average detection

accuracy, which was 9%, 4.7%, 0.9%, 2.3%, and 1.7% higher than that of Faster-RCNN, SSD, YOLOv5, YOLOv6, and YOLOv7, respectively. Additionally, the detection accuracy of AC-YOLOv5 exceeded 99%, meeting the requirements of industrial detection. There were also advantages in the detection of a single defect types, which were the best among the four networks.

**Table 7.** Test results of common networks in the fabric dataset.

| Method | AP | | | mAP | FPS |
|---|---|---|---|---|---|
| | L_line | S_line | Hole | | |
| Faster-RCNN | 91.3% | 83.5% | 95.4% | 90.1% | 31 |
| SSD | 93.6% | 92.4% | 97.1% | 94.4% | 86 |
| YOLOv5 | 96.4% | 98.6% | 99.5% | 98.2% | 476 |
| YOLOv6 | 98.2% | 96.6% | 95.6% | 96.8% | 405 |
| YOLOv7 | 97.7% | 96.7% | 97.9% | 97.4% | 250 |
| AC-YOLOv5 | 98.9% | 99% | 99.5% | 99.1% | 476 |

The method presented in this paper randomly selected three graphs with different defects to test each network model, and the experimental results are shown in Figure 8. Compared to the YOLOv5 model, the AC-YOLOv5 model improved the detection accuracy for all three types of defects, and the AC-YOLOv5 model also showed better performance compared to the other current mainstream networks.

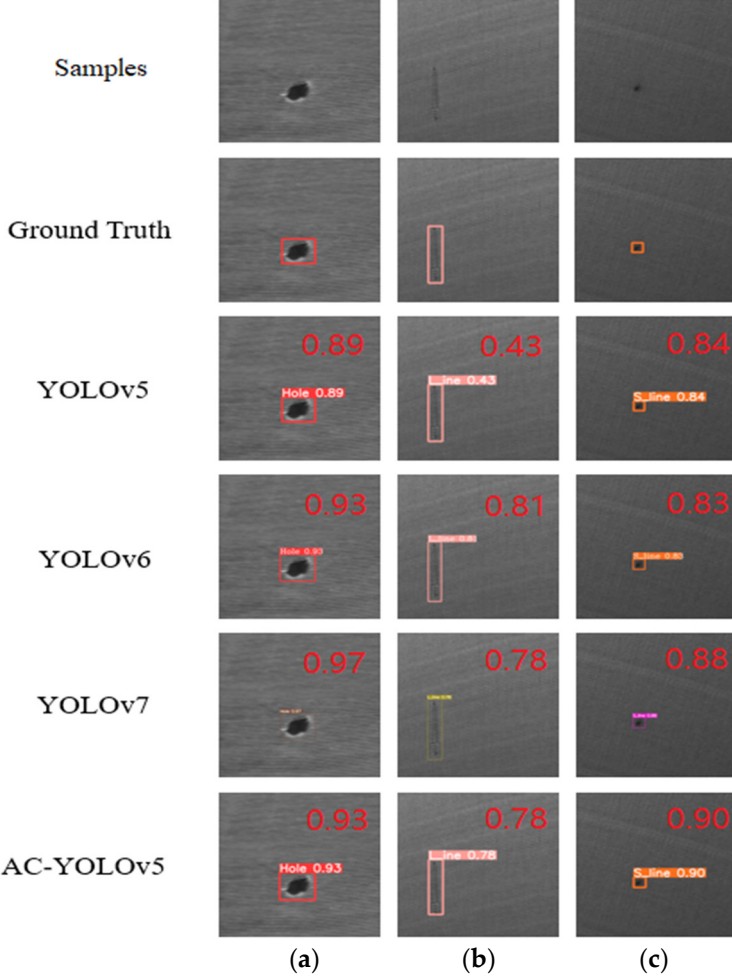

**Figure 8.** Test results of common networks on fabric datasets. (**a**) Hole, (**b**) L_ line, (**c**) S_ line.

*5.6. Experimental Results of the Light Guide Plate Dataset*

To further verify the validity of the AC-YOLOv5 model, this paper presents experiments carried out on the hot pressure light guide plate dataset (hot-pressed LGP) [37]. The experimental results are shown in Table 8, which shows that, compared to YOLOv5, AC-YOLOv5 improved the detection accuracy by 1.7%, 0.5%, and 1.1% for white, bright, and dark lines, respectively, and by 0.7% overall. Moreover, it reduced the complexity of the network and improved the detection speed.

**Table 8.** Test results of the hot-pressed LGP.

| Method | AP | | | | mAP | FPS |
|---|---|---|---|---|---|---|
| | **White Point** | **Bright Line** | **Dark Line** | **Area** | | |
| Faster-RCNN | 55.0% | 95.0% | 87.1% | 97.2% | 83.5% | 20 |
| SSD | 96.0% | 100.0% | 96.3% | 99.0% | 97.7% | 71 |
| YOLOv3 | 67.0% | 93.0% | 64.0% | 42.0% | 66.6% | 35 |
| YOLOv5 | 96.4% | 98.9% | 96% | 99.4% | 97.7% | 625 |
| AC-YOLOv5 | 98.1% | 99.4% | 97.1% | 98.9% | 98.4% | 555 |

## 6. Discussion

AC-YOLOv5 demonstrated the following advantages in this study:

(1) By adding the ASPP module to the backbone network, AC-YOLOv5 effectively integrated multiscale feature maps and processed objects of different sizes at the same time. It improved the receptive field and obtained feature maps with rich multi-level feature expression without loss of resolution, which further improved the feature extraction capability.

(2) The CSE module increased the weight of important features, increased the learning of target information, reduced the weight of unimportant features, and ignored some interference information, making the network more focused on defect recognition and effectively improving the defect detection ability.

(3) The experiment showed that the mAP of AC-YOLOv5 was improved by 0.9%. For L_line and S_line, the improvement was 2.5% and 0.4%, respectively, validating the effectiveness of the model.

There are still some weaknesses in this study and future research directions:

During training, our network model obtained rich feature information without losing the resolution of the image. However, when the resolution of the collected images was not sufficiently clear, the details and features of the targets in the images were easily blurred, leading to missing and false detections. Furthermore, our model was used for real-time detection of industrial fabrics. Industrial applications tend to favor lighter models; due to fabric deformation during production, the shape and appearance of defect detection may change, which may lead to a decrease in the accuracy of the model.

To address these potential limitations, future research could consider incorporating a regularization term into the network loss function to enhance the robustness of the model. In addition, model pruning can be applied to reduce the size and computational complexity of the network, with a focus on minimizing the impact on the detection performance.

## 7. Summary

In this study, a textile defect detection method based on AC-YOLOV5 was proposed to address complex textural backgrounds, different sizes, and types of textile defects. The proposed method first introduces an ASPP module in the backbone network to achieve multiple feature extraction, which facilitates the fusion of neck features and the acquisition of more feature information. Secondly, the CSE module was incorporated to analyze the importance of channel information, highlight defect information, and ignore background noise to enhance the detection accuracy. In addition, we collected a large number of textile images via a detection system set up in an industrial environment and established a textile

defect dataset for extensive experimental validation. The experimental results showed that the proposed method achieved detection accuracies of 99.5%, 98.9%, and 99% for hole, L_line, and S_line defects, respectively, and 99.1% overall, indicating its suitability for practical applications in the industrial sector.

**Author Contributions:** Conceptualization, J.L. and Y.G.; methodology, J.L.; software, X.K.; validation, X.K., Y.G. and Y.Y.; formal analysis, J.L.; investigation, J.L.; resources, J.L.; data curation, Y.Y.; writing—original draft preparation, Y.G.; writing—review and editing, J.L.; visualization, X.K.; supervision, J.L.; project administration, J.L.; funding acquisition, J.L. All authors have read and agreed to the published version of the manuscript.

**Funding:** This work was supported by the Key R&D Program of Zhejiang (No. 2023C01062), Basic Public Welfare Research Program of Zhejiang Province (No. LGF22F030001, No. LGG19F03001), and Guangdong Provincial Key Laboratory of Manufacturing Equipment Digitization (2020B1212060014).

**Data Availability Statement:** Availability of data and material—all data used in the experiments are from the self-built dataset. The datasets generated during the current study are available from the corresponding author upon reasonable request. The codes generated during the current study are available from the corresponding author upon reasonable request.

**Conflicts of Interest:** The authors declare no conflict of interest.

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
