# Peer review of "Automatic Fabric Defect Detection Method Using AC-YOLOv5"

_electronics, doi:10.3390/electronics12132950_

Round 1

Reviewer 1 Report

In this paper, the Authors present the improvement of YOLOv5 network to detect fabric defects more accurately than the base method. The overall quality of the paper is very good. The experiments are designed and presented correctly. I do not have any remarks. The only ones are related to typo mistakes, like "the resolution is 400 × 400 images" (line 306), here and there.

The English is satisfactory.

Author Response

Reviewer #1, Concern # 1: In this paper, the Authors present the improvement of YOLOv5 network to detect fabric defects more accurately than the base method. The overall quality of the paper is very good. The experiments are designed and presented correctly. I do not have any remarks. The only ones are related to typo mistakes, like "the resolution is 400 × 400 images" (line 306), here and there.

Author response: Thank you for your valuable comment on our paper. This is our problem. We have already made a revision of the entire article on word usage and grammatical issues.

Reviewer 2 Report

In this article, the authors proposed an AC-YOLOv5 model to detect fabric defects. They integrated YOLOv5 and atrous spatial pyramid pooling to effectively detect the fabric defect. The presented idea worthy therefore I recommend some major suggestions to enhance the quality of the paper.

1.      In the abstract limitation of the existing approaches are not highlighted and motivation of the proposed approach is not given. I recommend revising the abstract by adding a sentence, covering limitation of existing systems and motivation for the proposed system.

2.      The introduction section doesn’t follow the standard rules of manuscript writing. The authors need to revise the introduction section by incorporating the following points:

a.       Motivation: general (why the problem is important)

b.      Problem statement

c.       Current challenges

3.      Literature review section is not well-structured and contains coverage of old literature.

4.      The author needs to highlight their main contributions in bullets not in paragraph and also revise the contributions.

5.      Authors should the deeply revised conclusion section and discuss the limitations of the proposed method and their possible solutions in the future work.

6.      Reference needs to update.

https://www.sciencedirect.com/science/article/pii/S0952197623003573

https://www.sciencedirect.com/science/article/pii/S0306457323000262

I suggest to carefully review the English or use an English language editing service for that matter.

Author Response

Reviewer #2, Concern: Comments and Suggestions for Authors:

In this article, the authors proposed an AC-YOLOv5 model to detect fabric defects. They integrated YOLOv5 and atrous spatial pyramid pooling to effectively detect the fabric defect. The presented idea worthy therefore I recommend some major suggestions to enhance the quality of the paper.

Reviewer #2, Concern #1: In the abstract limitation of the existing approaches are not highlighted and motivation of the proposed approach is not given. I recommend revising the abstract by adding a sentence, covering limitation of existing systems and motivation for the proposed system.

Author response: Thank you for your valuable comment on our paper.We summarized the limitations of existing network models in the first section of our abstract, and proposed our own network model to address these limitations.

Reviewer #2, Concern # 2: The introduction section doesnt follow the standard rules of manuscript writing. The authors need to revise the introduction section by incorporating the following points:

  1. Motivation: general (why the problem is important)
  2. Problem statement
  3. Current challenges

Author response: Thank you for your valuable comment on our paper. This is our problem. We have rephrased this section.

Reviewer #2, Concern # 3: Literature review section is not well-structured and contains coverage of old literature.

Author response: Thank you for your valuable comment on our paper. We have rephrased the literature review section.

Reviewer #2, Concern # 4:The author needs to highlight their main contributions in bullets not in paragraph and also revise the contributions.

Author response: Thank you for your valuable comment on our paper. We have added a header in chapter one (1.1 The primary contributions of this study are as follows) and described the main contributions of this study in this section.

Reviewer #2, Concern # 5:Authors should the deeply revised conclusion section and discuss the limitations of the proposed method and their possible solutions in the future work.

Author response: Thank you for your valuable comment on our paper. We have revised the conclusion section and added a new chapter (6. Discussion). In this section, we presented the advantages of the AC-YOLOv5 model and discussed its limitations. We also proposed future research directions and improvement methods.

Reviewer #2, Concern #6: Reference needs to update.

Author response: Thank you for your valuable comment on our paper. We have added two references to the 10th and 11th positions of the reference list.

Reviewer #2, Concern #7:Comments on the Quality of English Language

I suggest to carefully review the English or use an English language editing service for that matter.

Author response: Thank you for your valuable comment on our paper. This is our issue. We have reviewed and revised the grammar throughout the entire article.

Reviewer 3 Report

Detection of fabric defect using AC-YOLO5 is proposed in this paper.

The paper is written and organized well. I have some questins regarding the proposed approach in this paper as follows :

- Have you tried to carry out comparison with the other methods rather than only YOLO methods?

-  Accuracy (in Table 8) is lower than YOLOv5 in case of "Area". What is your thought about this result?

-  Is the image resolution affecting the detection accuracy?

- Related work needs to be more specific.

Author Response

Reviewer #3, Concern:Comments and Suggestions for Authors

Detection of fabric defect using AC-YOLO5 is proposed in this paper.

The paper is written and organized well. I have some questins regarding the proposed approach in this paper as follows :

Reviewer #3, Concern #1: Have you tried to carry out comparison with the other methods rather than only YOLO methods?

Author response: Thank you for your valuable comment on our paper. We added the detection results of two currently popular object detection networks, SSD and Faster-RCNN, to Table 7 of the experimental comparison section. As compared, the YOLO series of network models have a significant advantage.

Reviewer #3, Concern #2:Accuracy (in Table 8) is lower than YOLOv5 in case of "Area". What is your thought about this result?

Author response: Thank you for your valuable comment on our paper.The overall accuracy of the model can be evaluated using metrics such as mAP, while the discriminative performance between different categories is reflected in the Precision-Recall curve. Under normal circumstances, there is an unavoidable relationship between Precision and Recall, that is, improving one indicator will inevitably affect the other. In practical applications, we need to choose appropriate thresholds or algorithms according to specific scenarios to balance the Precision and Recall metrics to achieve the best performance and effectiveness. In Table 8, although the detection accuracy of the Area defect exceeds the industrial requirements, other types of defects do not meet the requirements. Therefore, it is acceptable in industry to sacrifice some detection accuracy for Area defects to ensure that the detection performance of all defect types meets industrial requirements.

Reviewer #3, Concern #3:Is the image resolution affecting the detection accuracy?

Author response: Thank you for your valuable comment on our paper. Low image resolution will affect the detection accuracy. This is discussed in the limitation section of the supplementary chapter of the article, where the limitations of the proposed method are explained. In low-resolution images, the details and feature information of the targets are blurred or missing, which affects the accuracy of detection.

Reviewer #3, Concern #3:Related work needs to be more specific.

Author response: Thank you for your valuable comment on our paper. We have re-described the related work section and added shortcomings at the end of each research.

Round 2

Reviewer 2 Report

Most of my concerns were addressed in the revised version. The overall quality of this paper is improved.